# Review of Metasurfaces and Metadevices: Advantages of Different Materials and Fabrications

**DOI:** 10.3390/nano12121973

**Published:** 2022-06-08

**Authors:** Wei-Lun Hsu, Yen-Chun Chen, Shang Ping Yeh, Qiu-Chun Zeng, Yao-Wei Huang, Chih-Ming Wang

**Affiliations:** 1Department of Optics and Photonics, National Central University, Taoyuan 32001, Taiwan; ross6320@gmail.com (W.-L.H.); alau48@g.ncu.edu.tw (Y.-C.C.); 110286001@cc.ncu.edu.tw (S.P.Y.); qczeng@g.ncu.edu.tw (Q.-C.Z.); 2Department of Photonics, National Yang Ming Chiao Tung University, Hsinchu 30010, Taiwan

**Keywords:** metasurfaces, metalens, metadevices, aspect ratio, efficiency

## Abstract

Flat optics, metasurfaces, metalenses, and related materials promise novel on-demand light modulation within ultrathin layers at wavelength scale, enabling a plethora of next-generation optical devices, also known as metadevices. Metadevices designed with different materials have been proposed and demonstrated for different applications, and the mass production of metadevices is necessary for metadevices to enter the consumer electronics market. However, metadevice manufacturing processes are mainly based on electron beam lithography, which exhibits low productivity and high costs for mass production. Therefore, processes compatible with standard complementary metal–oxide–semiconductor manufacturing techniques that feature high productivity, such as *i*-line stepper and nanoimprint lithography, have received considerable attention. This paper provides a review of current metasurfaces and metadevices with a focus on materials and manufacturing processes. We also provide an analysis of the relationship between the aspect ratio and efficiency of different materials.

## 1. Introduction

Metamaterials that are classified as artificial materials and have subwavelength-scale structures possess novel light manipulation properties that do not exist in nature, such as negative-index media [1,2], optical cloaking [3,4], and super-resolution images [5,6,7]. These materials have attracted considerable attention over the past two decades, and these impressive optical phenomena that metamaterials exhibit are connected to the localized surface plasmon resonances (LSPRs) [8] that arise from the collective oscillations of free electrons in subwavelength metallic meta-units. Furthermore, multistacking metallic layers are occasionally introduced to enhance modulation [9]. However, metallic metamaterials suffer from Ohmic heat loss [10], which is further aggravated in multistacking metallic metamaterials. Moreover, multistacking metallic layers require several steps of lithography in sequence, which results in fabrication challenges [11,12,13,14]. To address these disadvantages, metasurfaces (single-layer metamaterials) have been proposed [15]. Metallic metasurface technology focuses on manipulating electromagnetic responses [16,17,18,19,20,21], including phase, amplitude, frequency, and polarization, without manipulating macroscopic artificial material properties. Versatile applications have been developed, such as metaholograms [22,23] and beam steering [24]. However, the Ohmic heat loss of metallic meta-units caused by the collective oscillation of free electrons at resonance condition limits the efficiency. Resonant and nonresonant dielectric metasurfaces with extremely low loss have been proposed to further enhance efficiency [25,26,27,28,29]. 

In 1690, Huygens proposed that each point on a wave front acts a secondary source of outgoing waves [30]. This principle become a well-known concept in electromagnetics and was termed Huygens’ principle. Employing the concept of Huygens’ principle, Huygens’ metasurfaces provide constructive interference of electric and magnetic responses at the operating wavelength (usually narrow bandwidth), resulting in unitary transmission as well as phase shifting [31]. Huygens’ metasurfaces were first demonstrated with metal antennas in the microwave region [32,33]. Then, a series of Huygens’ metasurfaces based on dielectric Mie scattering in the near-infrared region were proposed [32,34,35,36,37]. Although electric response and magnetic response only provide π phase shift across the spectrum, the two responses can be shifted together at the same wavelength by simply adjusting the geometric parameters of the nanostructure. The aspect ratio (the ratio of a structure’s thickness and its critical feature size) of dielectric Huygens’ metasurfaces is usually smaller than 1, which allows them to be easily nanofabricated. Dielectric Huygens’ metasurfaces require the refractive-index matching of top-covered material and substrate to satisfy the condition of constructive interference and maintain unitary transmission, which might limit some applications.

Propagation phase and geometric phase are widely used to engineer the phase distribution of dielectric metadevices. The propagation phase is referred to the optical path difference (OPD), as light propagates in the different effective refractive index of phase shifters [38,39,40,41,42]. According to effective medium theory, the OPD can be controlled by tuning the filling ratio of the material. On the other hand, the mechanism of the geometric phase, also known as Pancharatnam–Berry (PB) phase, was mainly established and generalized by Pancharatnam and Berry [43,44,45,46,47]. Briefly, circular polarized light converts its handedness when passing through a phase retarder (e.g., structured birefringence from an optical anisotropic structure), accumulating additional phase shift determined by the geometric orientation of the retarder. [48,49,50,51]. The arbitrary phase profile can be achieved by spatially varying geometric orientation of the optical anisotropic structure. Furthermore, based on geometric phase with dielectric nanopillars, dielectric metasurfaces exhibit no inherent Ohmic heat loss, possess high efficiency, and have been utilized for metalenses [52,53], metaholograms [54,55], metacorrectors [56], and related devices.

Metalenses, one popular application of flat optics based on dielectric metasurfaces, promise to replace conventional bulky refractive lenses [57,58]. In 1998, Lalanne et al. proposed blazed gratings with efficiencies greater than 80% at relatively large diffraction angles and preliminary results on lenses [59,60]. Subsequently, the Capasso group demonstrated the first metalens, a flat lens based on metasurfaces that works within a continual bandwidth of colors [61]. Soon, the metalens attracted considerable research attention. The phase distribution of a metalens renders it similar to a Fresnel lens, but it performs with higher efficiency because of the finer phase profile from pixelized meta-units and zero shadow area at Fresnel lens [62]. The concept of the metalens has been proposed several times over the years. An increasing number of studies demonstrate that the metalens has excellent potential for real-life consumer devices, for example, phone cameras [63,64]. However, current manufacturing processes and materials have yet to meet market requirements for cost of efficiency.

This paper reviews current metalenses and metadevices with a focus on their materials and manufacturing processes. We primarily focus on high-index dielectric materials for the visible range and their corresponding aspect ratios. Figure 1 provides an overview of the material classes, aspect ratios, and efficiencies that are described in detail throughout the paper. Aspect ratio is defined as the ratio of a structure’s thickness and its critical feature size. Many definitions of efficiency are used in different literatures. Polarization conversion efficiency (PCE) is defined by the ratio of cross-polarization power normalized to the total transmissive power, utilized in geometric phase. The efficiency of a phase distribution is usually discussed separately. For metalens applications, focusing efficiency is defined as the ratio of focal intensity to incident light intensity. For metaholograms, efficiency is usually defined as the ratio of pattern intensity to incident light intensity [65,66] or total power at the observing plane [67,68]. The efficiencies illustrated in Figure 1 are those claimed by the authors. Overall, Si-based [crystalline silicon (c-Si) and amorphous silicon (a-Si)] metadevices exhibit absorption loss in the visible range and are usually applied in the infrared of the spectrum. Si-disks are common in Huygens’ metasurfaces because of their higher refractive index (usually >3) and lower aspect ratio (usually <1). TiO_2_ and nitride-based (GaN and SiN_x_) metadevices offer a favorable refractive index and low absorption in the visible range. Mature light-emitting diode (LED) and complementary metal–oxide–semiconductor (CMOS)-compatible manufacturing techniques provide high-quality GaN and SiN_x_ films on substrates at low cost. These make nitride-based materials a great option for metadevice application. Ultraviolet (UV)-curable polymer resin (PER) is a common material used in the nanoimprint lithography (NIL) process. PER exhibits a low refractive index but can be enhanced by doping with high-refractive-index materials. Recently, perovskite metasurfaces have attracted much research interest because of their high refractive index, unique structural phase change, huge chirality, etc. However, the poor stability of perovskites creates a barrier to commercialization. Therefore, we will not discuss perovskite metasurfaces in detail [69,70,71,72]. In this paper, emerging material platforms for plasmonic and dielectric metasurfaces are discussed in Section 2. Fabrication methods are discussed in Section 3. Finally, a general discussion is provided in Section 4.

## 2. Metasurfaces Based on Different Materials

### 2.1. Metal-Based Metasurfaces

Metal-based metasurfaces manipulate the near-field and far-field radiation from the oscillations of free electrons at desired resonators—for example, the magnetic resonance from split-ring resonators (SRRs) [84,85,86,87,88,89,90] and electric resonance from nanorod response [91,92,93,94]. The scattering light features polarization, phase, amplitude, and momentum that are artificially coded by controlling the geometric parameters of metallic structures [95,96,97,98,99,100]. In 2011, Yu et al. proposed a wavefront-shaping metasurface to reconstruct a light beam by using a supercell consisting of eight different gold V-shaped antennas. The diagram of the V-shaped antenna is illustrated in Figure 2a [101]. Full 2π phase modulation can be achieved by changing the incline angle and orientation of the V-shaped antenna. The corresponding scattered electric field of the supercell under normal incidence at the x–z plane is also illustrated in Figure 2a. According to the Huygens–Fresnel principle, the solid red line can be regarded as the reconstructed wavefront of scattered light from V-antennas. Furthermore, the transmission propagates with an oblique angle that is called the anomaly diffraction and can be well-predicted by the generalized Snell’s law, which is a statement of momentum conservation at a phase gradient surface.

In addition to the deflection of a light beam, deflection of a more complex beam—for example, a holographic image—can also be achieved. Different metals feature considerably different plasma frequencies and working wavelength ranges. In 2015, Huang et al. proposed a multicolor metahologram to reconstruct colorful images in the visible range using aluminum (Al) plasmonic metasurfaces [102]. Figure 2b presents a schematic of the multicolor metahologram, which reconstructs images of “R” in red, “G” in green, and “B” in blue under linear polarized white light. Aluminum has a higher plasma frequency than gold and silver, allowing broadband plasmon resonances and applications in the visible to UV spectral range. Subwavelength-scale Al nanorods are deposited on a SiO_2_ layer over an aluminum mirror to generate a magnetic resonance (or gap resonance) that provides a 2π phase shift. The SiO_2_ spacer layer is necessary for manipulating the coupling strength between nanorods and Al mirrors to widen the phase modulation and reduce the Ohmic loss. However, due to the unavoidable loss of plasmonic resonance, the diffraction efficiencies for B, G, and R of this metahologram are 0.59%, 0.54%, and 0.69%, respectively, even when the SiO_2_ spacer layer has an optimized thickness.

Since each unit cell of a metasurface can be regarded as a scattering source and can be designed independently, it is possible to create a pixelized surface with an arbitrarily engineerable complex amplitude and propagation direction. An optical invisibility skin cloak is a fascinating example that eliminates the scattering from a rough surface by covering a metasurface. As illustrated in Figure 2c, Ni et al. experimentally demonstrated that an object covered by a metasurface skin cloak becomes invisible at a wavelength of 730 nm [103]. The skin cloak is an ultrathin layer consisting of subwavelength-scale gold nanorods, an MgF_2_ spacer layer, and a gold mirror. Plasmonic resonance of gold nanorods on the cloak surface contains the conjugate phase of a rough object and mimics the planar-reflecting wavefront so that the rough object can be hidden.

Plasmonic gap resonators integrated with active materials (e.g., semiconductors) are available to form electrically tunable metasurfaces, which have attracted considerable interest in recent years [104,105,106,107,108,109]. In 2021, Park et al. fabricated a three-dimensional (3D) LiDAR device based on an active metasurface composed of a gold nanoantenna array with electrically tunable channels, as illustrated in Figure 2d [110]. The resonance and phase shift of nanoantennas can be controlled by tuning the bias voltage. In principle, this can generate a continuous sweep within a hemisphere for 3D LiDAR applications. As illustrated in Figure 2d, a laser pulse is deflected by metasurfaces and illuminates the target objects. A synchronized photodetector detects the scattering signal from the objects, and the corresponding distances can be retrieved.

Although metal-based metasurfaces fabricated using LSPRs have been proven to exhibit impressive light modulation abilities, they are also accompanied by tremendous Ohmic heat loss. Therefore, efficiency improvements have become a key focus of metasurface development.

### 2.2. Si-Based Metasurfaces

Many experimentally created metallic metadevices exhibit significant Ohmic losses due to the inclusion of metal, resulting in lower efficiency. This presents a barrier to integration into optical devices. Except for Mie resonance [111,112,113,114], dielectric metasurfaces with propagation phase [19,115] and geometric phase [48,54,116] operate with neither resonance nor electron oscillation. Therefore, dielectric metasurfaces avoid losses and have attracted more interest from scientists and industry.

To support extensive phase coverage and low resonance loss, the materials used for dielectric metasurfaces need to have a high refractive index (n) and low extinction coefficient (k). The primary materials used for dielectric metasurfaces include Si, GaN, TiO_2_, and SiN_x_. Si is an abundant element and can be easily processed using standard CMOS-compatible manufacturing techniques [117,118,119,120,121]. Therefore, Si-based metasurfaces are regarded as low-cost metasurface platforms.

Yu et al. experimentally demonstrated a single-layer gradient metasurface with light regulation ability at visible wavelengths in 2015 [122]. This metasurface consisted of an a-Si nanodisk array on a fused silica substrate. The schematic of the metasurface is shown in Figure 3a. This all-dielectric metasurface was patterned using periodic unit cells. Each unit consisted of eight nanodisks of varying radii. The period and height of each nanodisk were fixed at 360 nm and 130 nm, and the radius varied from 120 nm to 155 nm (with a step of 5 nm) to obtain the desired phase modulation. Yu et al. also numerically demonstrated a nanodisk with electric and magnetic dipole resonance under incident wavelength at 715 nm, which allows for 2π phase control of the incident light. Compared with the propagation-phase metasurface, these unit cells do not require a high aspect ratio to achieve 2π phase control, which makes the fabrication process less challenging. Although a low aspect ratio provides advantages in manufacturing, the dipole resonance causes energy loss and presents a barrier to achieving high efficiency.

Metasurfaces that generate optical vortexes (also called optical angular momentum) are frequently utilized for quantum information and quantum computing [123,124]. Shalaev et al. experimentally demonstrated a Si-based metasurface with full 0-to-2π phase control at the NIR wavelength in 2015 [125]. The schematic phase distribution of the metasurface is illustrated in Figure 3b. This metasurface approached the optical vortex by increasing the phase of the nanoblocks in a clockwise direction. Phase modulation was achieved by controlling the electric and magnetic resonance with different dimensions of nanoblocks.

As illustrated in Figure 3c, Khorasaninejad et al. demonstrated a grating to redirect impinging light based on its handedness in 2014 [126]. This grating consisted of an a-Si nanofin on a glass substrate. The supercell of the grating contained six nanofins that feature azimuthal asymmetry. To achieve high efficiency, the height of the nanofin must correspond to a waveplate. Figure 3c presents the scanning electron microscopy (SEM) images of the grating. The nanofin had a width of 50 nm, a length of 300 nm, and a height of 1500 nm. The aspect ratio of the nanofin reached 30. For left circular polarization illumination, the measured m = −1 order’s power reached approximately 76%. The m = 1 order’s power approached zero. This grating exhibited a high extinction ratio.

Tunable lenses consisting of a pair of adjacent optical elements are widely used to control optical power. Current tunable lenses, including Alvarez–Lohmann lenses [127] and moiré lenses [128,129], are bulky. Therefore, tunable metalenses are attracting a growing amount of attention. Tunable metalenses are classified into two categories based on their operating mechanism: inherent and external. Inherent tunable metalenses are commonly created by adjusting the position or the shape of the building blocks. Polydimethylsiloxane (PDMS) is one of the most widely used stretchable materials [130,131,132,133,134,135]. PDMS is used for generating a reconfigurable metalens. External mechanical forces are used to adjust the position of the building blocks. Kamali et al. used PDMS to make a tunable lens with focal distance tunability from 600 µm to 1400 µm in 2016, as illustrated in Figure 3d [136].

External tunability involves controlling the responses of a metalens or a metalens assembly to external stimuli without changing the functions of the metalens. For example, Fan et al. switched the focal length of a metalens between two focal planes by simply changing the linear polarizations of the incident light, as illustrated in Figure 3e [137]. The other example is integrating a metalens into a micro-electro-mechanical system (MEMS). Arbabi et al. demonstrated a MEMS-integrated metalens system in 2018, as shown in Figure 3f [138]. Electrostatic MEMS platforms provide mechanical stress for lens deformation to achieve focus-tunable optical elements. To make metadevices tunable on a MEMS system, the metadevices are fabricated on a flexible substrate, such as a membrane or PDMS [139,140,141]. Moreover, in addition to the mechanically tunable metadevices mentioned above, tunable metasurfaces based on liquid crystal are an alternative. The orientation of the anisotropic liquid crystals can be both electrically and thermally controlled. The optical properties of a metasurface can be thus modified [142,143,144,145]. Although the efficiency is presently not favorable, the technique exhibits promise.

### 2.3. TiO_2_-Based Metasurfaces

Although many metadevices have been created using Si-based metasurfaces, they are not optically transparent, resulting in limited applications in the visible range. Therefore, the development of high-efficiency metasurfaces shifted focus to TiO_2_ in 2016 [78,146], with work mainly done by the Capasso group. With a high refractive index and low optical loss, TiO_2_-based metasurfaces exhibit exceptional properties in the visible wavelength range [147,148,149,150,151,152,153,154,155,156,157,158,159,160,161,162,163]. 

Khorasaninejad et al. demonstrated a high-aspect-ratio and high-efficiency TiO_2_-based metalens with NA = 0.8, operating wavelengths of 405, 532, and 660 nm, and a phase shift based on the PB phase (Figure 4a) [78]. Three TiO_2_ nanofin structures with varying widths of 40–85 nm and lengths of 150–410 nm and a uniform thickness of 600 nm were the building blocks of the metalens. The PCE of meta-atoms can reach up to 86% and vary with different operation wavelengths. The efficiency of metadevices also depends on their functionality. For a metalens, focusing efficiency can be defined as the power of the focal spot normalized to the incident power. The efficiency of metadevices varies depending on function. The focusing efficiency of a metalens is up to 73%. The same group also characterized their metalens by using light sources with undesigned wavelengths and chromatic aberrations. This triggered further study of achromatic metalenses in the visible spectrum.

The Capasso group subsequently experimentally demonstrated several types of achromatic metalenses with TiO_2_-based meta-atoms [79,164,165,166]. The structured dispersion and material dispersion of meta-atoms were considered. The group delay and group delay dispersion of meta-atoms and their precise location on an achromatic metalens were analyzed [164]. Chen et al. made a 400-nm period and aimed at a 470–670 nm achromatic lens. In their 2018 work, the gap of the two nanofins was fixed at 60 nm for term position modulation [79]. Figure 4b indicates the focal length of different metalenses produced through different types of dispersion engineering. The measurement results are illustrated in Figure 4b: n = 0 for the achromatic lens, n = 1 for the diffraction lens, and n = 2 for the stronger dispersion lens.

Figure 4c,d [77,167] provides examples of the multifunctionality of a TiO_2_ metasurface. Mueller et al. used both the geometric and propagation phases to design a hologram and used dual channels in arbitrary orthogonal polarization states. More channels of the incident polarization were changed to produce two patterns. Hu et al. used three channels to produce a trichromatic holographic film. Through the optimization of different wavelengths and polarizations, wavelength multiplexing was demonstrated. 

Generally, atomic layer deposition (ALD) is used to produce higher-quality TiO_2_. Compared with chemical vapor deposition and physical vapor deposition, namely sputtering, the ALD process is time-consuming. It is not widely used in the semiconductor process and is unsuitable for mass production.

### 2.4. GaN-Based Metasurfaces

To address the issue of the fabrication challenges of TiO_2_-based metasurfaces as well as the Ohmic heat loss of metallic metasurfaces and absorption loss of Si-based metasurfaces, the Tsai group first proposed GaN-based metasurfaces [73]. GaN is a direct bandgap III-V material with a bandgap of 3.4 eV (equal to a wavelength of approximately 364.67 nm). No intrinsic absorption exists throughout the visible spectrum. Therefore, a high-quality GaN provides a favorable platform for applications of visible metasurfaces [168,169,170,171,172], owing to its relatively high refractive index and high transparency. Furthermore, enabled by mature LED process technology, a high-quality GaN on sapphire substrate [173,174,175,176,177,178] is an excellent low-cost option for metasurface applications.

In 2017, Emani et al. experimentally demonstrated a wide-range and high-efficiency polarization beam splitter by using GaN-based metasurfaces [81]. As demonstrated on Figure 5a’s left side, epitaxial GaN film deposited on a sapphire substrate for the following fabrication process. The GaN metasurface was patterned using electron-beam lithography (EBL) and an inductively coupled reactive ion etching process. The highest critical aspect ratio of the nanostructure observed was approximately 5.75. Figure 5a presents the energy distribution in various diffraction orders of metasurfaces with p-polarized light. Incident light was deflected into the T + 1 order at the operating wavelength of 460 nm. The experimental device efficiency was 73%. 

With a focus on color display technology in recent years, more attention has been given to the ability of metasurfaces to reshape the light distribution in the visible range. Song et al. demonstrated a polarization-multiplexing hologram based on a PB-phase metasurface in 2020 [54]. Figure 5b presents the schematic of a multidirectional metahologram. Four holographic phase profiles were encoded into four different subpixel arrays to generate four polarization images: the letters C, N, R, and S. Generally, the meta-atoms are arranged periodically, which causes a grating effect and generates a diffraction pattern with multiple diffraction orders. To eliminate such ghost images produced by the grating effect, the random arrangement of meta-atoms is necessary. The inset of Figure 5b presents the SEM image of meta-atoms in four different arrangements that can effectively eliminate the ghost images. 

Metalenses designed in visible light have exhibited great potential for application and commercialization. Chen et al. experimentally demonstrated a GaN-based metalens with an off-axis focusing ability as a color router based on wavelength multiplexing in 2017, as shown in Figure 5c [73]. The metalens deflected as well as focused three primary colors into desired spatial positions with a high operating efficiency up to 91.6% for visible light. SEM images are presented in the inset of Figure 5c. Their work demonstrated that metadevices with a color multiplexing or multifunctionality arrangement of meta-atoms possess high potential in CMOS image sensor applications. 

In the same research group, a GaN-based metalens array was applied to the problem of optical quantum information technology. Li et al. proposed a stable and controllable platform for quantum optical source research in 2020 [74]. A 10 × 10 metalens array is illustrated in Figure 5d. The metalens was composed of GaN nanopillars with a height of 800 nm, period of 200 nm, and various diameters. By using the metalens array and beta barium borate nonlinear crystal, they successfully demonstrated 2-, 3-, and 4D two-photon path entanglement with phases that encoded fidelities of 98.4%, 96.6%, and 95.0%, respectively.

Lin et al. used a GaN achromatic metalens array to experimentally demonstrate a full-color light-field imaging system. They successfully created a chromatic-aberration-free full-color light-field camera in 2019 [63]. The metalens array consisted of 60 × 60 lenses, and every single achromatic metalens consisted of over 9000 integrated resonant unit elements. A schematic of the full-color light-field imaging system and rendered images are presented in Figure 5e. Reconstructed images with different depths of focus were rendered from the subimages on the sensing plane by using the achromatic metalens array with circularly polarized light.

### 2.5. SiN-Based Metasurfaces

The SiN_x_ material system, primarily in the form of SiN_x_, remains the subject of intense research and manufacturing interest across multiple technological fields. The refractive index of SiN_x_ is approximately 2.03, which is slightly lower than that of GaN (2.29). However, SiN_x_ is more conducive to CMOS processing [179,180]. This characteristic gives SiN-based metasurfaces the potential for mass production.

In 2016, Zhan et al. experimentally demonstrated a low-contrast and high-quality metalens based on SiN_x_ [75]. They simulated different conditions of periodicity (p), substrate thickness (t), and the diameter of nanopillars by using rigorous coupled-wave analysis. Owing to the relatively low refractive index of SiN_x_, a higher aspect ratio of meta-atoms is required to improve efficiency. In this work, the aspect ratio of the optimal design was 6.6, which presented a fabrication challenge. The optimized conditions were *t* = λ and *p* = 0.7 λ, as illustrated in Figure 6a. This metalens featured an N.A. of 0.75 operating in the visible range. The focusing efficiency was as high as 40%.

In the same research group, Fan et al. also demonstrated a SiN-based metalens with a large N.A. (0.98) and diameter (1 cm) in 2018, as illustrated in Figure 6b [181]. The transmission was approximately 80% for nonpolarized light in the visible range. A submicron focal spot was realized, and the focusing efficiency was approximately 40%.

Zhan et al. also presented a design method for freeform optics using a SiN-based metasurface operating in the visible range [182]. The aim of freeform optics research is to design optical elements with more complex phase geometries beyond rotational symmetry to enhance performance and minimize size, resulting in many complex and asymmetric curvatures. An example is the Alvarez lens [139], which is an aberration-correcting lens with adjustable focus. However, the complex forms and high curvatures of freeform optics present a manufacturing challenge. Therefore, metasurfaces are introduced to convert freeform optics by transferring the complex geometric curvatures to a flat surface with wavelength-scale thickness to facilitate manufacturing. The phase-sag profile of the freeform surface is quantized to six linear steps from 0 to 2π. The corresponding cylindrical nanoposts are capable of producing a full cycle of phase shifts while maintaining large regions of continuous and near-unity transmission amplitude. Zhan et al. fabricated a cubic metasurface and a corresponding square Alvarez metasurface with area of 150 μm × 150 μm in 633 nm SiN_x_ film deposited on a fused quartz substrate. SEM images of the finished devices coated in gold are presented in Figure 6c. The point spread functions (PSFs) of the cubic metasurface and the PSFs of Alvarez metasurface are presented in the bottom part of Figure 6c. These metasurfaces offered a broadband operation. Therefore, the PSFs’ patterns were similar under red and green illumination.

In conclusion, SiN-based metasurfaces exhibit excellent optical-control performance with high efficiency in the broadband spectrum, including the visible and UV range. In addition, the CMOS compatibility of SiN_x_ makes it more attractive for large-scale manufacturing. However, silicon-rich SiN_x_ films suffer from significant optical loss in the blue spectral range. Hydrogenated silicon nitride (SiN_x_:H) has relatively high transparency in the full visible range but a relatively low refractive index. It offers a trade-off between transparency and refractive index.

## 3. Fabrication Methods

### 3.1. UV Lithography

The primary semiconductor manufacturing method is based on pattern-repeatable lithography processes, such as stepper [76,183,184,185] and scanner [186,187] processes. Compared with EBL, photolithography is more suited for use in mass production. As a product of the maturation of semiconductor fabrication technology, stable and cheap commercial electronics have proliferated, completely changing human civilization and life. As noted, the excellent light control properties of metasurfaces have received extensive attention in recent years. Metasurfaces (such as metalenses and waveplates) are likely to become emerging optical components in consumer products. Due to the tiny feature size of meta-atoms, patterning using the EBL process is the most common fabrication method. However, EBL suffers from low production efficiency and high production costs. Therefore, producing metasurfaces in large quantities at low cost has become a research priority.

UV-light step-and-repeat cameras (steppers) are the most popular lithography equipment used in CMOS manufacturing. Although steppers have the advantages of relatively low cost and high productivity, their spatial resolution is limited by the diffraction limit. By contrast, the critical dimensions of meta-atoms for the visible range are in the hundred-nanometer scale, coinciding with the minimum linewidth of a commercial *i*-line stepper.

At present, deep UV lithography (DUVL) using a stepper and scanner platform is the most mature approach for mass-producing electronic devices and can also be used for metadevice mass production. She et al. fabricated large-area a-Si metalenses by using photolithographic stepper technology in 2018. A schematic of metalens production is presented in Figure 7a [185]. The fabrication process is as follows: the a-Si, photoresist, and contrast enhancement material are deposited on a wafer substrate. The pattern of the metalens is defined by the *i*-line stepper photolithography process. The pattern is etched and transferred to the a-Si layer by using reactive ion etching. Multiple metalens elements can be obtained after dicing the substrate. Photographic and SEM images of an a-Si metalens are displayed in the inset of Figure 7a. The fabrication processes are highly compatible with existing semiconductor process technologies and provide a good demonstration of the mass production of large-area metasurfaces.

Electronic tunable technologies are always a goal in the development of electronic devices. Mass-produced metasurfaces combined with electronic tunable systems are a key potential application of metadevices. She et al. experimentally demonstrated an electrically tunable large-area metalens in 2018 [183]. This tunable device is integrated with an a-Si-based metalens and dielectric elastomer actuators (DEAs), as illustrated in Figure 7b. The simple production process is as follows: a-Si patterns are fabricated on a GeO_2_ layer on a silicon wafer by using the aforementioned UV lithography method. The patterns then are transferred from the silicon wafer to a membrane of the elastomer, and the scaffold membrane is bonded to a DEA. The thickness of this device is only 30 μm. The focal length of the device is controlled by DEA using five addressable electrodes. Figure 7b reveals the measurement intensity profile along the z-axis. When a voltage bias is applied, the focal length of the metalens increases significantly from 50.1 mm to 53.1 mm. This demonstrates the advantages of metalens and electronic control components and provides a feasible example of electronic control metadevices.

Park et al. successfully created an all-glass metalens at the centimeter scale that features focusing and imaging capabilities at visible wavelengths. Figure 7c presents a photograph of 45 metalenses on a 4-inch glass wafer. The SiO_2_ nanopillars of the metalenses were patterned using a DUVL stepper and dry-etching process; a SEM image is shown in Figure 7c [80]. The minimum diameter of the nanopillars is 250 nm, with a height of 2 µm and a sidewall angle of 2.85°. With such a high aspect ratio of the structure, maintaining the verticality of the side wall during the dry-etching process is highly challenging. In 2016, Khorasaninejad et al. numerically demonstrated that a 1-nm difference in the radius of the nanopillars causes a 3° phase shift in light modulation [146]. The dimensional deviation also induces doubt regarding the light modulation ability of the metasurface. However, Figure 7c shows that a slight dimensional deviation will not affect the operation of the metalens. Compared with a commercial aspheric lens and plano-convex lens, a metalens exhibits aberration restraint and a superior focusing profile.

### 3.2. Nanoimprint Lithography

NIL technology has the advantages of large-area patterning and low cost and is a solution to achieving high throughput and resolution [188,189,190,191,192,193,194]. The first step of NIL requires a patterned master mold usually defined by the EBL process. Once a mold is made, the replication process of NIL is much faster than that of EBL. In addition, NIL is also compatible with the current fabrication process of the panel industry. Therefore, NIL dramatically increases the possibility that metasurface devices can be commercially mass-produced. Recently, many demonstrations of NIL have required recoating the mold with films as ink before the replication process. However, the films residual on the mold required additional removal procedures or reduced the number of usages of the mold. This procedure also reduced the productivity. Removing this procedure in the manufacturing process has become a focal challenge in developing NIL technology.

Lee et al. experimentally demonstrated a see-through metalens with a diameter of 20 mm fabricated using NIL in 2018 [195]. A photograph of the metalens that was produced to downsize the lens set in an augmented reality (AR) headset is displayed in Figure 8a. This metalens consists of poly-Si nanostructures on a quartz substrate. Figure 8a also presents a SEM image of the nanostructures and AR images. Due to its large-area metalens, the AR near-eye display has the advantages of an ultrawide field of view, full visible color, and high resolution. In the fabrication of the metalens, the pattern of a silicon master mold was first defined by using the standard EBL process. Several films (Au, Cr, and SiO_2_ patterned as a hard mask) were then evaporated on the master mold as ink [195]. The hard mask pattern was then transferred to the poly-Si film for the next etching process. The poly-Si metalens was ready after the removal of the hard mask residue. This method applies NIL technology to the production of semiconductor-based lenses. However, the recoating of several films is necessary for every process, and the residues limit the lifetime of the mold. 

To further increase the lifetime of the mold and improve the productivity of the NIL process, Kim et al. developed a facile nanocasting method to fabricate a dielectric metahologram in 2019 [82]. Figure 8b presents a schematic of the NIL process. First, a nanostructured master mold is made using EBL followed by the etching process. The pattern is then transferred to hard-PDMS (h-PDMS) by using thermal curing. Next, the dielectric particle-embedded UV-curable PER is spin-coated on the PDMS polymer mold to replicate its nanostructures, and this is followed by transfer to a glass substrate. This method only requires PER film recoating on the mold, which can increase the lifetime of the mold and improve productivity. Figure 8b presents the simulated holographic image and the reconstructed image under blue (λ = 450 nm), green (λ = 532 nm), and red (λ = 635 nm) laser illumination. The characteristics of low cost and favorable fabrication productivity mean there is considerable potential for development, although the highest efficiency of the demonstrated hologram was 46%.

NIL technology can quickly transfer the pattern of the metasurface to a UV-curable resin at low cost. However, the refractive index of the typical resin is so low that it limits the performance of the metasurface. Therefore, in 2020, Yoon et al. added TiO_2_ nanoparticles to the resin to increase the equivalent refractive index [83]. The equivalent refractive index of the resin can be determined by the proportion of mixed nanoparticles. The schematic of the fabrication process is shown in Figure 8c, with SEM images corresponding to each fabrication stage. Optical micrographs are inset in the SEM images. The fabrication processes are described as follows. The pattern of master mold was first defined by the EBL process. Subsequently, h-PDMS and PDMS were sequentially coated on the master mold to transfer the pattern and act as a buffer layer. The composite of UV-curable resin and TiO_2_ nanoparticles was dropped on a glass substrate to prepare for mold imprinting. Finally, by illuminating UV light on the imprinted UV-curable resin, a solid pattern was obtained. This method provides a straightforward solution to increase the equivalent refractive index of the resin.

### 3.3. Other Processes

In addition to the known DUV lithography and nanoimprint process, other methods have the potential for deployment in the mass production of metasurfaces. Thus, this paper next discusses several promising studies of manufacturing methods to fabricate large-area metasurfaces at low cost.

Template-encoded microlens projection lithography (TEMPL) was first proposed by Gonidec et al. [196] to overcome one of the challenges of metasurface fabrication: the rapid prototyping of patterns with similarly shaped structures. This method consists of two main steps, as shown in Figure 9a. First, a microlens projection lithography system is fabricated by arranging self-assembled arrays of spheres on PDMS. Second, the photomask pattern is projected onto the substrate. Compared with the photomask, the projected image is significantly reduced by approximately 10,000 times in size in a single step, which reduces the resolution requirement of the photomask to values >1 mm. This characteristic enables the photomask to be fabricated more easily by using conventional printing or laser cutting. The qualities of low cost, simple production, and successful fabrication of large-area metasurfaces make TEMPL a promising candidate for future research.

For conventional metasurface fabrication, defining a pattern by using a lithography process is necessary. Recently, Nemiroski et al. proposed a shadow-sphere lithography process to fabricate a nanoantenna [197]. This process can define metasurface patterns in a large area without using any lithography process. A schematic of the shadow-sphere lithography process is shown in Figure 9b. Multiple polystyrene spheres with 1-μm diameter are deposited on the surface of a silicon wafer. The size of polystyrene spheres can be precisely controlled by adjusting the etching time of oxygen plasma etching. The evaporation materials are deposited at the gaps between the spheres during the evaporation process and define the metasurface pattern. The pattern can be controlled by changing the tilt angle of deposition. The authors demonstrated multiple material nanorods with a hexagonal arrangement, as displayed in Figure 9b. Usually, a process of several steps of lithography in sequence is necessary to fabricate this kind of complex structure. Shadow-sphere lithography provides a new strategy for the efficient prototyping and discovery of periodic metasurfaces made with multiple materials. Although this technique is limited in the design of patterns, it is still a good option for the large-area fabrication of metasurfaces.

The next process can fabricate metasurfaces by using a high-energy laser rather than using traditional CMOS process equipment. Figure 9c presents a photograph of a large-area (8 mm × 8 mm) glass-based metasurface fabricated by Zhou et al. [198] The pattern of the metasurface was directly defined on a glass substrate by using a femtosecond pulse laser. The silicon glass substrate was decomposed into porous glass through high-energy laser exposure, and the refractive index of glass was determined by laser intensity. Figure 9c displays the polariscope optical image of the metasurface sample mark in Figure 8c. SEM images are shown in the inset of Figure 9c. The dimension of nanostructures is on the order of approximately 30–100 nm, which matches the subwavelength condition for visible-light applications.

## 4. Conclusions

Dielectric-based metasurfaces modulate the phase of incident electromagnetic waves by using the bound charge resonance, Mie resonance, propagation phase, geometric phase, and other methods. Compared with metallic-based metasurfaces, the overall efficiency is relatively high due to the low Ohmic loss. However, the thickness of dielectric-based metasurfaces is usually 10 to 100 times thicker than that of metallic-based ones. Therefore, a fabrication technique for high-aspect-ratio structures, which are incredibly challenging to fabricate with a CMOS-compatible process, is needed. This section summarizes the performance specifications of several dielectric metasurfaces listed in Table 1. Here, the aspect ratio is calculated as the thickness divided by the critical dimension (CD). Although a series of studies have demonstrated that metasurfaces with a small CD and a high aspect ratio can be fabricated using EBL and deep dry etching, these processes are expensive and challenging for mass production. In addition to the choice of materials, aspect ratio dramatically affects the difficulty of mass production. Making a metasurface with a high aspect ratio is more challenging. However, for metadevices with relatively low aspect ratios, optical efficiency is occasionally sacrificed, especially for metasurfaces based on propagation-phase modulation. Therefore, the efficiency and aspect ratio should be simultaneously considered. As noted, the refractive index of Si-based (c-Si and a-Si) materials is 3–4. Therefore, an aspect ratio smaller than 5 is sufficient for efficient operations in the near-infrared spectral range. However, Si-based metasurfaces suffer from considerable absorption loss in the visible range. Both nitride-based (GaN and SiN_x_) and TiO_2_-based metasurfaces offer a fair refractive index and low absorption, making them suitable for visible applications. The indices of GaN and SiN_x_ are 2.4 and 2, respectively. An aspect ratio as high as 10–15 is needed for GaN-based and SiN-based metasurfaces to obtain sufficient phase modulation for favorable efficiency. SiN-based metasurfaces should have a higher aspect ratio than GaN-based ones. For the same aspect ratio, the efficiency of SiN_x_ is lower than that of GaN. This phenomenon is clearer for oxide-based materials. The refractive index of TiO_2_ is much higher than that of SiO_2_. Therefore, SiO_2_ requires a much higher aspect ratio to improve light manipulation capabilities and efficiency. Moreover, the refractive index of TiO_2_ depends highly on the crystalline phase. The refractive index of rutile, anatase, and amorphous TiO_2_ varies from 2.6 to 1.8. Although crystalline TiO_2_ has a high refractive index, the difficulty of controlling crystalline phase limits the application. In the case of PER, the effective refractive index is improved by doping with TiO_2_ particles. Consequently, PER doping with TiO_2_ particles simultaneously reduces the aspect ratio requirements and enhances efficiency. As noted, a low aspect ratio benefits mass production but reduces efficiency. Increasing the material’s refractive index is an effective way to simultaneously achieve a low aspect ratio and high efficiency, but it is not suitable for every material. Striking a balance between aspect ratio and efficiency is something to which metasurface designers must continue to attend.

The dimensions of meta-atoms for the visible range are in the scale of hundreds of nanometers. Therefore, high-resolution EBL is the main method to fabricate meta-atoms for academic purposes. However, the high cost and low productivity pose a challenge to the commercialization of metasurfaces. UV light steppers are the most popular lithography equipment in CMOS manufacturing due to their lower cost and high productivity. Unfortunately, their spatial resolution is too large to define the tiny critical dimension of visible metasurfaces. Therefore, an UV light stepper is only suitable for near-infrared-range applications. A NIL utilizes a high-resolution master mold fabricated by the EBL process. Although the cost of the master mold is high, the replication process of NIL is much faster than that of EBL. Moreover, NIL has the advantages of large-area patterning. Combining the above advantages, NIL is an alternative for mass production of metasurfaces and metadevices.

## Figures and Tables

**Figure 1 nanomaterials-12-01973-f001:**
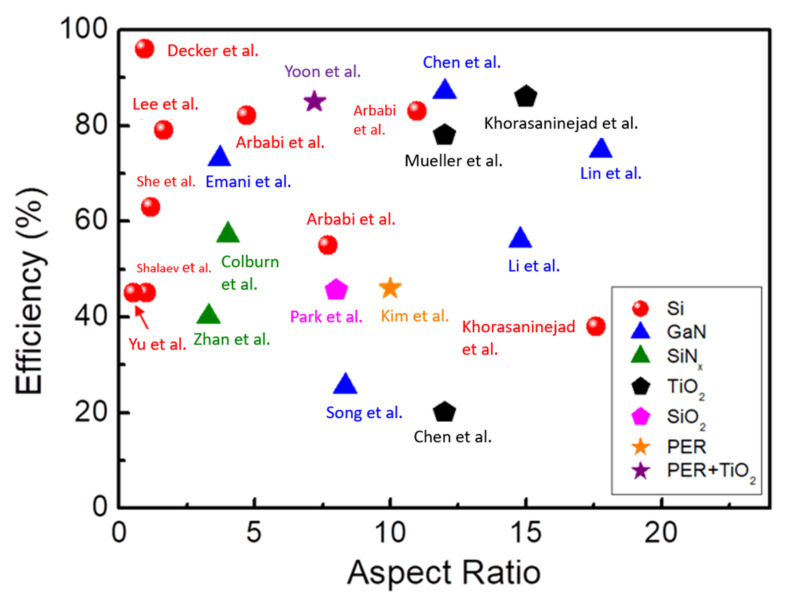
Aspect ratio versus efficiency of metasurfaces based on different materials. The materials silicon (Si) [37,73,74,75,76,77,78,79,80], gallium nitride (GaN) [54,63,73,74,81], silicon nitride (SiN_x_) [75,76], titanium oxide (TiO_2_) [77,78,79], silicon oxide (SiO_2_) [80], polymer resin (PER) [82], and PER doping with TiO_2_ [83] are marked with red dots, blue triangles, green triangles, black pentagons, pink pentagon, orange star, and purple stars, respectively.

**Figure 2 nanomaterials-12-01973-f002:**
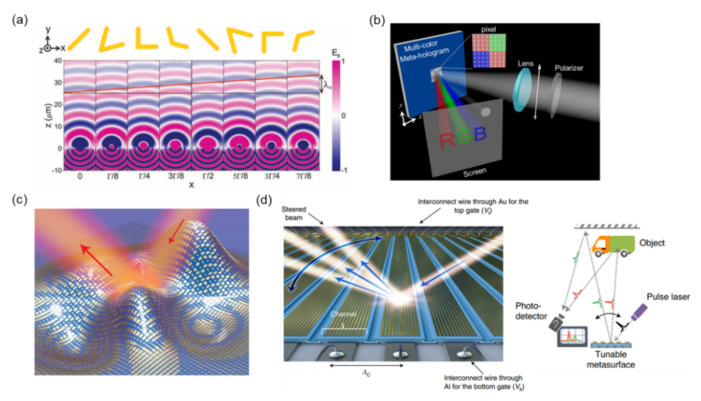
(**a**) Finite-difference time-domain simulations of the scattered electric field with V-antenna arrays [101]. (**b**) Illustration of the multicolor metahologram [102]. (**c**) Schematic of a metasurface skin cloak [103]. (**d**) Illustration of 3D LiDAR composed of a metasurface and electrically tunable channels [110].

**Figure 3 nanomaterials-12-01973-f003:**
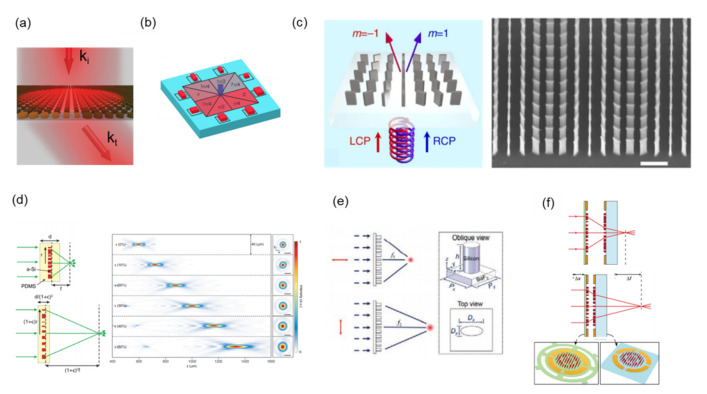
(**a**) Beam deflection metasurface [122]. (**b**) Phase distribution of an optical vortex metasurface [125]. (**c**) Schematic (left) of a-Si grating that redirects impinging light based on its handedness; SEM image (right) of a-Si grating with a scale bar of 1 µm [126]. (**d**) Schematic of a dielectric metalens with a tunable focal distance (left); measured optical intensity profiles of a tunable metalens (right) [136]. (**e**) Polarization-controlled focusing metalens [137]. (**f**) Tunable lens that consists of a stationary lens on a substrate and a moving lens on a membrane [138].

**Figure 4 nanomaterials-12-01973-f004:**
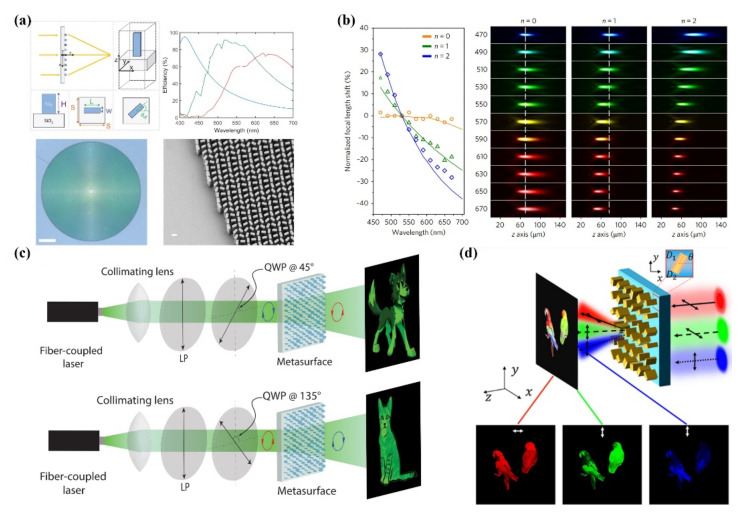
(**a**) TiO_2_ metalens. The unit cell is a nanofin (top left); its PCE is illustrated in the top right [78]. (**b**) Experimentally measured focal length for TiO_2_ achromatic metalenses [79]. (**c**) Polarization multiplex metahologram [77]. (**d**) Trichromatic metahologram [167].

**Figure 5 nanomaterials-12-01973-f005:**
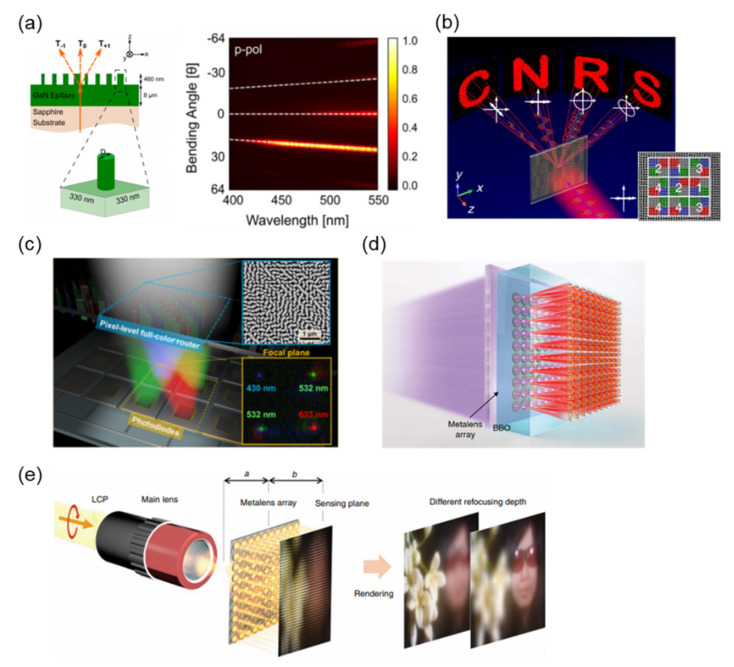
(**a**) Schematic of a metasurface (left); measured energy distribution of the metasurface illuminated by the p-polarized light (right) [81]. (**b**) Multidirectional metahologram; SEM image is shown in the inset [54]. (**c**) Multicolor router with a dielectric metalens in a CMOS image sensor [73]. (**d**) Quantum light source based on a metalens array [74]. (**e**) Light-field imaging application with a metalens array [63].

**Figure 6 nanomaterials-12-01973-f006:**
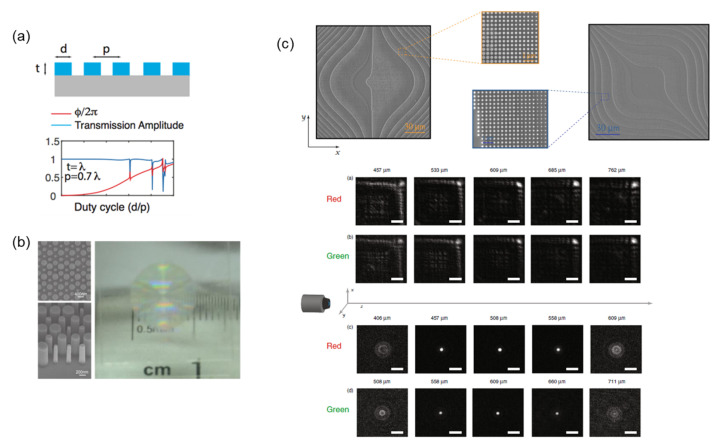
(**a**) Grating structures (top) with periodicity p can be formed by using cylindrical posts (with diameter d) arranged in a square lattice. The thickness of the grating is denoted as t. The thickness t was λ, and periodicity p was 0.7 λ [75]. (**b**) Top-view SEM image (left) of a portion of the fabricated micrometalens with the grating constructed using a low-refractive-index SiN_x_ metalens with circular nanopillars on a fused silica substrate. The hexagonal lattice constant is a = 416 nm, and the thickness of nanopillars is 695 nm. Photograph (right) of the macroscopic metalens with a focal length of −4 mm [181]. (**c**) SEM images of half of the Alvarez lens (upper left), the cubic phase plate coated in gold (upper right), PSFs of the cubic element under coherent illumination using red and green light (top middle), and PSFs of a 500-μm metasurface lens under red and green illumination (bottom middle) [182].

**Figure 7 nanomaterials-12-01973-f007:**
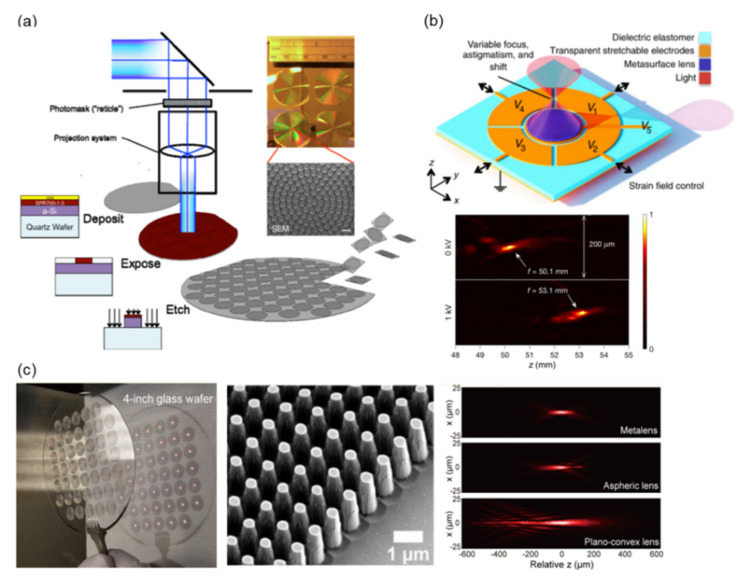
(**a**). Metalens production using photolithographic stepper technology [185]. (**b**) Tunable metalens device (top), in which the focal length of the metalens is controlled using DEAs; intensity profile (bottom) along the *z*-axis with and without voltage bias [183]. (**c**) A photograph (left) and SEM images (middle) of metalenses on a 4-inch glass wafer. Comparison of measured light intensity distributions (right) of the focal spot of a fabricated metalens and commercial plano-convex lens, respectively [80].

**Figure 8 nanomaterials-12-01973-f008:**
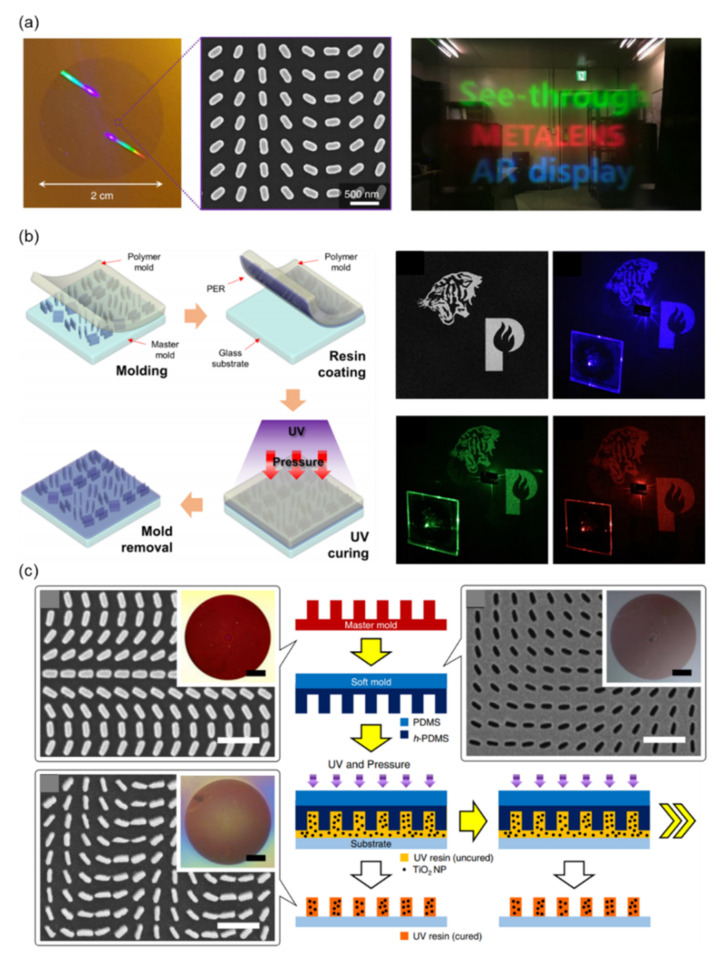
(**a**). Photograph of a see-through metalens with a diameter of 20 mm (left); a SEM image of part of the PB-phase metalens (middle); AR images in red, green, and blue (right) [195]. (**b**) Schematic of the nanocasting process (left) and simulated holographic image and measured holographic image under incident lasers of λ = 450, 532, and 635 nm (right) [82]. (**c**) Schematic of the metalens fabrication process. SEM images and photographs (inset) are the master mold, soft mold, and final metalens. All scale bars of the SEM and photographs are 1 and 100 μm, respectively [83].

**Figure 9 nanomaterials-12-01973-f009:**
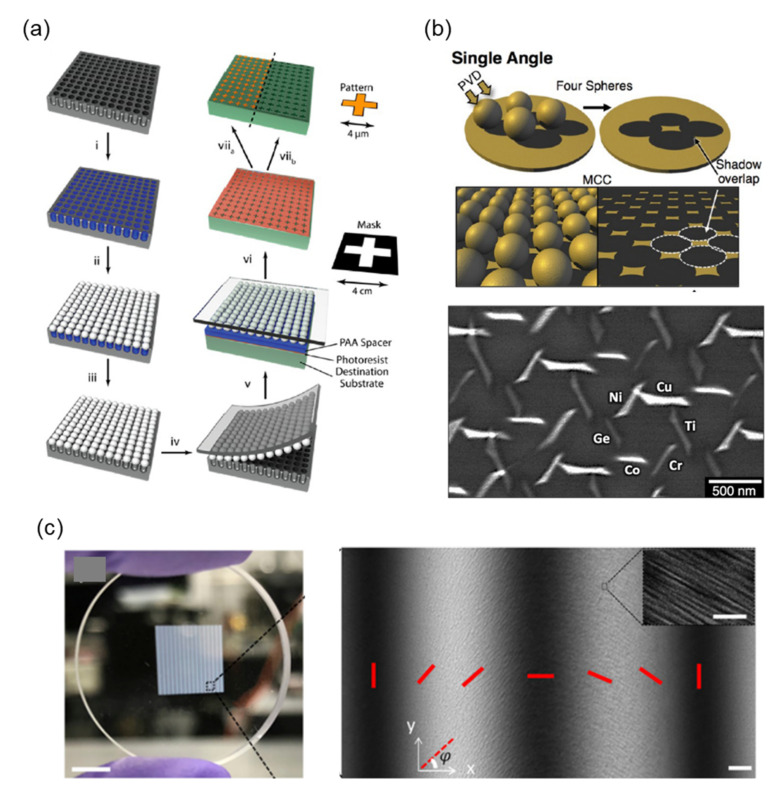
(**a**) Schematic of the TEMPL process [196]. (**b**) Schematic for deposition (top) of the shadow-sphere lithography process; SEM images (bottom) of multimaterial structures composed of 10 nm of either Cu, Ti, Cr, Co, Ge, or Ni [197]. (**c**) Photograph of a glass-based metasurface (left) with a pattern area of 8 mm × 8 mm; polariscope optical image (right) of the dotted square area of Figure 9c left. The red bars correspond to the orientation of the structures in one period (scale bar is 25 μm). The inset (right) is the SEM image with a scale bar of 500 nm [198].

**Table 1 nanomaterials-12-01973-t001:** Reported performance of dielectric-based metasurfaces.

Material	Wavelength (nm)	Aspect Ratio	Eff. (%)	Process	Application	Ref.
a-Si	1358	0.93	96%	EBL	Wavefront engineering	[36]
a-Si	1550	4.7	82%	EBL	Metalenses	[115]
a-Si	705	0.54	45%	EBL	Beam steering	[122]
a-Si	974	17.6	38% (m = 1)	EBL	Beam steering	[126]
a-Si	915	7.7	55%	EBL	Metalenses	[138]
a-Si	1550	1.17	63%	SL	Metalenses	[183]
a-Si	915	11	83%	EBL	Metalenses	[199]
poly-Si	1550	1	45%	EBL	Vortex converter	[125]
poly-Si	473–660	1.66	79%	NIL	Metalenses	[195]
GaN	600	8.33	25.40%	EBL	Metahologram	[54]
GaN	404	14.8	56%	EBL	Quantum source	[74]
GaN	430–470	3.7	73.00%	EBL	Beam splitting	[81]
GaN	430–633	12	87% @430 nm	EBL	Metalenses	[73]
GaN	400–660	17.77	74.8% @420 nm	EBL	Metalenses	[63]
SiN_x_	633	3.29	40%	EBL	Metalenses	[75]
SiN_x_	1550	4	57%	SL	Metalenses	[76]
TiO_2_	532	12	78%	EBL	Metahologram	[77]
TiO_2_	405–660	15	86% @405 nm	EBL	Metalenses	[78]
TiO_2_	470–670	12	20% @500 nm	EBL	Metalenses	[79]
SiO_2_	488–660	8	45.6% @633 nm	SL	Metalenses	[80]
PER	450–635	10	46% @532 nm	NIL	Metahologram	[82]
PER + TiO_2_ particle	450–635	7.2	85% @532 nm	NIL	Metalenses	[83]

EBL: electron-beam lithography; SL: stepper lithography; NIL: nanoimprint lithography; PER: polymer resin.

## Data Availability

Not applicable.

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
