# Peer review of "Review of Metasurfaces and Metadevices: Advantages of Different Materials and Fabrications"

_nanomaterials, 2022, doi:10.3390/nano12121973_

Round 1
Reviewer 1 Report
The authors present a comprehensive nicely written review of the recent developments in the field of optical metasurfaces, paying special attention to the materials and technologies used for their fabrication. The topic is a true research hotspot and definitely deserves a careful analysis. I believe that the paper will be of interest for the audience of Nanomaterials. However, I would like to draw the authors’ attention to the points they have missed, and also to a few technical issues.
Major:
1) To mark the current prospects of metasurface implementation in real-life consumer devices, one can check www.metalenz.com and possibly refer to it avoiding commercial advertising.
2) The authors briefly discuss on page 6 (lines 241-251) the value of tunable metasurfaces but only in relation to Si-based ones and only mechanically tunable. Note that recently a Si metalens has been successfully tuned by a liquid crystal: https://pubs.acs.org/doi/10.1021/acs.nanolett.1c00356 Also, a TiO2 metasurface has been also successfully tuned by a liquid crystal: https://www.science.org/doi/10.1126/science.aaw6747 Similar tunability of a fused-silica metasurface has been also studied: http://www.pnas.org/lookup/doi/10.1073/pnas.2006336117 Pure liquid-crystal metasurfaces are also very conveniently tunable: https://pubs.acs.org/doi/10.1021/acsphotonics.0c01168
3) It is a pity that the authors totally skip the hot trend of metal halide perovskite metasurfaces. Please check: https://onlinelibrary.wiley.com/doi/10.1002/lpor.201900079 , https://pubs.acs.org/doi/10.1021/acs.nanolett.8b01912, https://www.nature.com/articles/s41467-022-29253-0 and also a nice brief review https://onlinelibrary.wiley.com/doi/10.1002/adom.201800784.
Technical:
- The sentence in lines 39-42 is confusing. It will be hard for a general reader to understand the causality here. Please reformulate.
- Please strictly define the aspect ratio before discussing it starting from lines 53-54.
- Note also that it is impossible to extract from Fig.1 which point corresponds to which paper. For instance, I was intrigued by the aspect ratio close to 20, but was unable to look further.
- The paragraphs in lines 221-230 and 231-240 are identical.
- Not all references are complete! I randomly checked and came across the missing data for [95,106,110,111,118,119,143,144,166,167,170].
Reviewer 2 Report
the paper is very interesting. The authors show many metasurface based on dielectric medium.
I have some comments.
in the table 1 , the range of the wavelength is of order of some nm .
It will be useful have another table for different range of wavelength.
In the same table, the authors should be define the efficiency.
Reviewer 3 Report
This manuscript provides review on the advantages of different materials and fabrication methods related to metasurfaces and metadevices. In particular, various metasurfaces built on the basis of various materials (including metals, Si, TiO2, GaN, and SiN) are systematically classified and representative research works in each field are introduced in a balanced way so that the readers in various fields could understand how metamaterials can be created and fabricated. Notably, contrary to the previous review papers reported, this review provides the introduction to a wide variety of both materials and fabrication methods, and thus it will attract broad interest in fields of nanoscience, materials science and photonics. However, I recommend some corrections before the manuscript can be accepted:
- Some new metadevices were overlooked by the authors in section 2.1, i.e.: Sci Rep 9, 20367 (2019); Nat Commun 10, 2118 (2019); Advanced Functional Materials 31.20 (2021): 2010329; Sci Rep 11, 1795 (2021).
- Some pictures lack the optical properties of metadevices.They only represent the concept of the device.
- In conclusion part, there is lack of perspective or insight in the field of metadevices/fabrications. Adding perspectives of each section could improve the quality of the manuscript.
